# A Deep Learning Approach for Motion Forecasting Using 4D OCT Data

**Marcel Bengs**[*1]                                                        MARCEL.BENGS@TUHH.DE

**Nils Gessert**[*1]                                                          NILS.GESSERT@TUHH.DE

**Alexander Schlaefer**[1]                                                SCHLAEFER@TUHH.DE

[1] *Institute of Medical Technology, Hamburg University of Technology, Hamburg, Germany*

## Abstract

Forecasting motion of a specific target object is a common problem for surgical interventions, e.g. for localization of a target region, guidance for surgical interventions, or motion compensation. Optical coherence tomography (OCT) is an imaging modality with a high spatial and temporal resolution. Recently, deep learning methods have shown promising performance for OCT-based motion estimation based on two volumetric images. We extend this approach and investigate whether using a time series of volumes enables motion forecasting. We propose 4D spatio-temporal deep learning for end-to-end motion forecasting and estimation using a stream of OCT volumes. We design and evaluate five different 3D and 4D deep learning methods using a tissue data set. Our best performing 4D method achieves motion forecasting with an overall average correlation coefficient of 97.41%, while also improving motion estimation performance by a factor of 2.5 compared to a previous 3D approach.

**Keywords:** 4D Deep Learning, Optical Coherence Tomography, Motion Estimation, Motion Forecasting

## 1. Introduction

Forecasting of patient or surgery tool movements is a relevant problem for various medical procedures ranging from radiotherapy (Ren et al., 2007) to ophthalmic interventions (Kocaoglu et al., 2014). Typically, motion tracking requires a fast imaging modality. Optical coherence tomography (OCT) is an imaging modality with a high spatial and temporal resolution, allowing for 4D real-time imaging (Wang et al., 2016). OCT has been integrated into intraoperative microscopes (Lankenau et al., 2007), and has also been considered for monitoring laser cochleostomy (Pau et al., 2008). Often, OCT applications operate on small-scale, delicate structures where motion can disrupt the workflow or even cause injury (Bergmeier et al., 2017), hence accurate motion tracking is particularly relevant, e.g., for field of view (FOV) adjustment during intraoperative imaging (Kraus et al., 2012) or for adjustment of surgery tools during automated interventions (Zhang et al., 2014). Recently, deep learning methods have shown promising results for motion estimation using two OCT volumes (Gessert et al., 2019; Laves et al., 2019), or for pose estimation of a marker object using a single OCT volume (Gessert et al., 2018). While these methods have inference times in the range of milliseconds (Gessert et al., 2019), performing the actual compensation or adjustment introduces a lag between the actual adjustment and the motion estimation. This

---

[*] Contributed equally

can be problematic if fast and large motions occur. One approach to overcome this problem is motion *forecasting* for predicting the future trajectory, which requires a time series, instead of just two volumes for estimating a motion vector (Gessert et al., 2019). Processing sequences of 3D volumes leads to the challenging problem of 4D deep learning, where immense computational and memory requirements make architecture design very difficult. In this work, we propose an end-to-end deep learning approach for motion estimation and forecasting using entire sequences of OCT volumes. So far, 4D deep learning has only been considered for few examples, e.g., in the context of functional magnetic resonance imaging (Bengs et al., 2019) and computed tomography (Clark and Badea, 2019). We evaluate several 4D deep learning methods using a tissue data set and demonstrate a new mixed 3D-4D deep learning approach.

## 2. Methods

**Deep Learning Methods.** We formulate a supervised learning problem where we predict the current motion vector $\Delta s_{t_n} \in \mathbb{R}^3$ as well as future motion vectors $\Delta s_{t_{n+1}} \in \mathbb{R}^3$, $\Delta s_{t_{n+2}} \in \mathbb{R}^3$ of a region of interest (ROI), given a 4D OCT image sequence $x_t = [x_{t_0}, x_{t_1}, ..., x_{t_n}]$ capturing the history of a trajectory. We employ a dense neural network (Huang et al., 2017) as a baseline, using 3 densenet blocks with 3 layers each, connected with average pooling layers. Before the final regression layer of the network with 9 outputs, we use global average pooling. Using this baseline network, we evaluate five methods for processing of the 4D data, shown in Figure 1. First, we only consider the initial and the last volume of a sequence processed by an architecture that has been proposed for motion compensation (Gessert et al., 2019), where two volumes are processed by a two-path CNN with shared weights. The outputs of the two-path CNN are concatenated into the feature dimension and then processed by our densenet baseline with 3D convolutions. (2-Path-CNN3D) Second, we use the entire sequence of volumes and extend the two path approach to a multi path architecture with shared weights, while the number of paths is equal to the number of input volumes. (n-Path-CNN3D) Third, we consider the entire sequence of volumes and directly learn from both spatial and temporal dimensions by using 4D spatio-temporal convolutions. We employ three 4D convolutional layers followed by our densenet with 4D convolutions. (CNN4D) Fourth, we use a mixed 3D-4D approach, by applying a multi-path architecture for individually processing each volume of a sequence. Then, we reassemble a temporal dimension by concatenating the outputs into a temporal dimension, and afterward we apply our baseline network with 4D spatio-temporal convolutions. (n-Path-CNN4D) Fifth, we consider a gated recurrent neural network with convolutional gating operations (Xingjian et al., 2015) in front of our 3D baseline CNN, similar to (Bengs et al., 2019). (GRU-CNN3D)

**Data Set.** We consider sequences of volumetric OCT images capturing motions of 40 different ROIs of a chicken breast sample, using a FOV of $5\,\text{mm} \times 5\,\text{mm} \times 3.5\,\text{mm}$ and a volume size of $32 \times 32 \times 32$ pixels. We use the setup proposed by Gessert et al. (2019) with a swept-source OCT device (OMES, OptoRes) and a robot. This setup employs a scanning stage with mirror galvanometers allowing to shift the OCT's laser beam and thus the FOV, which can be utilized for automated data set generation and annotation. Using this setup we acquire sequences of volumetric OCT data $x_t = [x_{t_0}, x_{t_1}, x_{t_2}, x_{t_3}, x_{t_4}]$ for 100 different

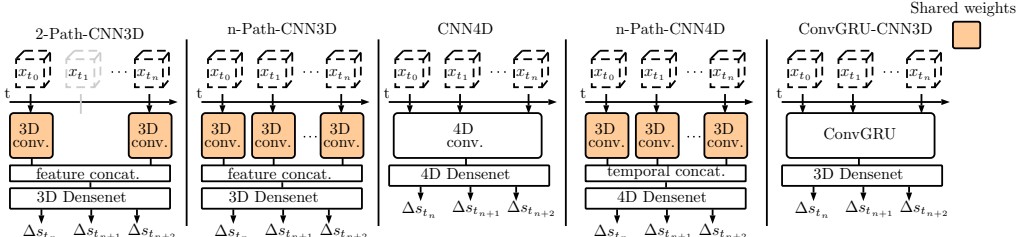

Figure 1: The five architectures we employ.

Table 1: Results for motion estimation ($\Delta s_{t_n}$) and forecasting ($\Delta s_{t_{n+1}}$, $\Delta s_{t_{n+2}}$). Errors are given in mm.

|  | MAE $\Delta s_{t_n}$ | MAE $\Delta s_{t_{n+1}}$ | MAE $\Delta s_{t_{n+2}}$ | aCC (%) | Inf. Time (ms) |
|---|---|---|---|---|---|
| 2-Path-CNN3D | $0.40 \pm 0.48$ | $0.48 \pm 0.61$ | $0.57 \pm 0.73$ | 77.49 | $3.61 \pm 0.37$ |
| n-Path-CNN3D | $0.25 \pm 0.29$ | $0.31 \pm 0.36$ | $0.37 \pm 0.44$ | 92.27 | $4.51 \pm 0.17$ |
| CNN4D | $0.19 \pm 0.21$ | $0.23 \pm 0.26$ | $0.28 \pm 0.31$ | 96.44 | $12.52 \pm 0.16$ |
| n-Path-CNN4D | $0.16 \pm 0.18$ | $0.19 \pm 0.22$ | $0.23 \pm 0.27$ | 97.41 | $12.00 \pm 0.12$ |
| GRU-CNN3D | $0.20 \pm 0.21$ | $0.25 \pm 0.26$ | $0.31 \pm 0.32$ | 95.49 | $12.76 \pm 0.17$ |

trajectories with corresponding labels $\Delta s_t = [\Delta s_{t_4}, \Delta s_{t_5}, \Delta s_{t_6}]$ for each of the 40 ROI. The ROIs have the same size as the OCT's FOV, hence at $t_0$ the FOV complete overlaps with a ROI. The label $\Delta s_{t_4}$ refers to the relative three dimensional motion vector between the initial position of a ROI at $t_0$ and the current one at $t_4$, while $\Delta s_{t_5}$ and $\Delta s_{t_6}$ are future relative motion vectors. For data generation, we consider various smooth curved trajectories with different motion magnitudes generated with spline functions. To evaluate our models on previously unseen ROIs, we randomly choose five independent ROIs for testing and validating and use the remaining 30 ROIs for training. We optimize our networks for 400 epochs using Adam.

## 3. Results and Discussion

We report the mean absolute error (MAE) for estimation and forecasting separately, also we consider the average correlation (aCC) coefficient for combined performance in Table 1. As expected, extending a previous approach (2-Path-CNN3D) to the processing of sequences (n-Path-CNN3D) already improves motion estimation and motion forecasting performance. However, comparing the different models for processing of the 4D data, n-Path-CNN3D performs worse. This indicates that temporal processing using the channel dimension is challenging, as already shown in the natural image domain (Tran et al., 2015). While learning from the 4D data with CNN4D and GRU-CNN3D performs well, combining the multi path approach with a 4D architecture increases performance even further, suggesting an effective spatial preprocessing. Moreover, there are no optimized, native 4D convolution operations available so far and yet our computational expensive 4D models still achieve

competitive motion estimates with up to 83 Hz. Overall, we demonstrate that 4D deep learning methods enable high accuracy motion forecasting based on time series of volumetric data.

## Acknowledgments

This work was partially funded by Forschungszentrum Medizintechnik Hamburg (grants 04fmthh16).

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
