# OpenReview forum: "A Deep Learning Approach for Motion Forecasting Using 4D OCT Data "
_MIDL.io/2020/Conference — MIDL 2020_

### Official Review · AnonReviewer1 · 2020-02-25
**Interesting work, but lack of detail in experimental setup**

**Rating:** 3
**Confidence:** 3

**Review:**

This paper evaluates 5 different models for motion tracking in 4D OCT. The models are variants of that proposed in Gessert et al (2019), which is here extended in different ways to perform motion forecasting/prediction using a sequence of OCT volumes, rather than motion estimation between 2 OCT volumes.

On the positive side, the extension of the Gessert model to motion forecasting seems like a useful one. The methods employed seem reasonable and quantitative evaluation is performed to compare them. The discussion of the results reveals findings that may well be of interest to others.

However, one weakness of the paper was that the details of the experimental setup for data generation were not clear without following up the Gessert et al (2019) reference. Was the setup the same as in Gessert et al (2019), i.e. with a robot moving the object and mirrors moving the OCT FOV? Please modify the paper to make this clear.

Also, can the authors comment on what the accuracy requirement is for motion tracking in OCT?

Other specific suggestions:
•	Section 2: “… region of interest (ROI) performing motions …” does not make sense to me. Maybe get rid of “performing motions”?
•	Section 2: In description of n-Path-CNN3D, “extent” should be “extend”
•	Section 2, Dataset: “For data generation, we consider various smooth curved trajectories with different motion magnitudes” – this is a bit vague, can you provide more information? How were these trajectories formed? How big were the ROIs?
•	Section 3: “combing” should be “combining”

---

### Official Review · AnonReviewer4 · 2020-03-01
**Deep Learning-based Motion forecasting in 4D OCT data - Review**

**Rating:** 3
**Confidence:** 3

**Review:**

The authors compare five different approaches for motion estimation and forecasting in a “chicken breast sample” OCT experiment. The main result is that taking into account the temporal nature of the data (eg. by 4D-convolution and especially using appropriate spatial preprocessing of the data) allows for improved motion estimation and prediction.

From the methodical perspective, the authors highlight the applied *4D* deep learning approaches, which indeed are still not commonly applied in the medical imaging domain. At this, it should be noted that their application is also not novel per se. However, the proposed joined spatial preprocessing of the individual OCT frames (inspired by the underlying publication Gessert et al, 2019) is an interesting idea.

In general, the manuscript is well structured and written. As "Cons" points: central aspects that are necessary to interpret  the results are not given (at least not in the manuscript; some are listed in the underlying paper by Gessert et al, 2019): temporal resolution, image resolution, motion span and velocity range of the trajectory. Furthermore, which are the actual OCT applications that are addressed, what are typical motion patterns and ranges – and what are corresponding application-driven requirements in terms of eg. MAE?

In summary: The contribution offers a nice comparison study of 3D vs. 4D approaches for deep learning-based motion forecasting. Due to missing information, it is, however, hard to interpret the results besides the obvious gain in motion estimation and prediction accuracy.

---

### Official Review · AnonReviewer3 · 2020-03-13
**A nice comparison of different approaches for deep learning on a time-series of 3D volumes**

**Rating:** 4
**Confidence:** 4

**Review:**

The authors present and evaluate five different methods for estimation of a motion vector from a series of 3D OCT volumes. The application is interesting and the proposed network architectures are intuitive and seem appropriate for the problem at hand. The use of 4D convolutions has not been explored extensively, and it is nice to see an application for them.

Some minor remarks:

- Please include the resolution (size in voxels as well as mm) of the input volumes.

- What do the 12 outputs of the network represent exactly? If I understand correctly, the ouputs are 3D motion vectors (3 numbers), times 3 time points $\Delta s_{t4}, \Delta s_{t5}, \Delta s_{t6}$. Would this not make 9 outputs?

- In Figure one, it seems that one of the outputs is $\Delta s_{t_0}$. Should this not be $\Delta s_{t_n}$?

- Data has been generated using smooth curved trajectories. It would be interesting to know how these were generated, and whether this resembles real data.

- It took me a while to relate the 83Hz in the text to the 12ms in the Table, maybe make this more explicit.

---

### Official Review · AnonReviewer2 · 2020-03-19
**A Deep Learning Approach for Motion Forecasting Using 4D OCT Data**

**Rating:** 3
**Confidence:** 5

**Review:**

This work presents a 4D spatio-temporal deep learning for end-to-end motion forecasting and estimation using a stream of OCT volumes. The proposed method is validated on OCT 3D sequences. Compared to the alternative 3D CNN strategies, this work shows 4D CNN achieve better motion estimation and forecasting results.

The motivation is clear. It is easy to follow the paper. Some implementation details are missing (eg. parameter settings). Overall, this short paper shows some promising results of using 4D CNN in motion analysis.

---

### Meta-Review · Area_Chair1 · 2020-04-03
**MetaReview of Paper156 by AreaChair1**

**Rating:** 4

**Metareview:**

All reviewers unanimously suggest acceptance of this paper that proposes novel methodological elements (in the context 4D deep learning) and also provides a good comparative evaluation. The small remaining concerns can easily be fixed, I therefore strongly recommend to accept the paper.

**Paper Type:**

both

---

### Decision · Program_Chairs · 2020-04-11

Accept